# In Silico Prediction and Prioritization of Novel Selective Antimicrobial Drug Targets in *Escherichia coli*

**DOI:** 10.3390/antibiotics10060632

**Published:** 2021-05-25

**Authors:** Frida Svanberg Frisinger, Bimal Jana, Stefano Donadio, Luca Guardabassi

**Affiliations:** 1Department of Veterinary and Animal Sciences, Faculty of Health and Medical Sciences, University of Copenhagen, DK 2000 Frederiksberg, Denmark; bimal@sund.ku.dk; 2Naicons Srl, 20139 Milan, Italy; stefano.donadio@ktedogen.com

**Keywords:** antimicrobial targets, *Escherichia coli*, in silico, microbiota

## Abstract

Novel antimicrobials interfering with pathogen-specific targets can minimize the risk of perturbations of the gut microbiota (dysbiosis) during therapy. We employed an in silico approach to identify essential proteins in *Escherichia coli* that are either absent or have low sequence identity in seven beneficial taxa of the gut microbiota: *Faecalibacterium*, *Prevotella*, *Ruminococcus*, *Bacteroides*, *Lactobacillus*, *Lachnospiraceae* and *Bifidobacterium*. We identified 36 essential proteins that are present in hyper-virulent *E. coli* ST131 and have low similarity (bitscore < 50 or identity < 30% and alignment length < 25%) to proteins in mammalian hosts and beneficial taxa. Of these, 35 are also present in *Klebsiella pneumoniae*. None of the proteins are targets of clinically used antibiotics, and 3D structure is available for 23 of them. Four proteins (LptD, LptE, LolB and BamD) are easily accessible as drug targets due to their location in the outer membrane, especially LptD, which contains extracellular domains. Our results indicate that it may be possible to selectively interfere with essential biological processes in *Enterobacteriaceae* that are absent or mediated by unrelated proteins in beneficial taxa residing in the gut. The identified targets can be used to discover antimicrobial drugs effective against these opportunistic pathogens with a decreased risk of causing dysbiosis.

## 1. Introduction

Due to the worldwide increase in resistance observed among certain bacterial pathogens, there is a pressing need for novel antimicrobials. Most of the antimicrobial drugs approved for human use since the end of the antibiotic golden age in the 1960s belong to known antimicrobial classes and are primarily active against Gram-positive bacteria [1]. Although in recent years novel antimicrobials have become available for effective treatment of Gram-negative bacterial infections, such drugs belong to known antimicrobial classes, and only one antimicrobial with a novel mechanism of action, Murepavadin, has made it to Phase 3 trials [2,3]. Thus, there is a demand for truly novel antimicrobial compounds targeting Gram-negative bacteria. In 2017, the World Health Organization released a list of global priority antimicrobial-resistant (AMR) pathogens in order to guide research, discovery and development of new antibiotics [4]. The highest category (‘Priority 1: Critical’) comprises *Acinetobacter baumannii, Pseudomonas aeruginosa* and several *Enterobacteriaceae*, including *Escherichia coli*. The latter species is the most frequent cause of several common infections, such as urinary tract infection and septicaemia, that are increasingly difficult to treat due to the global spread of specific hyper-virulent and multidrug-resistant clones. In particular, ST131 is a major contributor to the global spread of fluoroquinolone resistance and extended-spectrum β-lactamase-mediated resistance to β-lactams, and is responsible for millions of multidrug-resistant infections each year [5,6].

Oral antimicrobial therapy impacts the healthy gut microbiota by inducing a loss of beneficial microbes followed by expansion of opportunistic pathogenic bacteria, such as *Enterobacteriaceae* [7]. This phenomenon, generally referred to as dysbiosis, can in extreme cases lead to life-threatening secondary infections caused by *Clostridioides difficile* [8], which are becoming increasingly common and difficult to treat. Certain antimicrobial-sensitive taxa residing in the healthy gut microbiota, e.g., *Bacteroidetes* and *Lachnospiraceae*, have been shown to provide protection from *C. difficile* infections through colonization resistance [7]. Moreover, the gut-associated taxa *Lactobacillus* and *Bifidobacterium* both contain strains that are associated with beneficial effects on health and are used as probiotics in various food supplements, including *Lactobacillus planetarum, L. paracasei, L. acidophilus, Bifidobacterium infantis, B. longum* and *B. breve* [9].

Pathogen-targeted antimicrobial drugs with a limited effect on beneficial organisms could potentially decrease the risk of dysbiosis and antibiotic-induced secondary infections. One approach to discover such drugs is the employment of target-based assays to identify compounds that selectively interfere with the viability of *E. coli* and other *Enterobacteriaceae* without affecting the healthy gut microbiota. This requires identification of targets that are specific for this bacterial family. Here, we performed an in silico study to identify protein drug targets in *E. coli* that: (i) are present in hyper-virulent *E. coli* ST131; (ii) do not display significant similarity to host proteins; and iii) are absent or have low amino acid identity in selected members of the healthy gut microbiota. Furthermore, the selected proteins were analyzed for their conservation and essentiality in the closely related pathogen *K. pneumoniae*, and their ‘druggability’ was assessed based on subcellular localization (SCL), availability of three-dimensional structures and presence of known inhibitors.

## 2. Results

### 2.1. Essential Genes in the Target Pathogen

Due to the lack of understanding of the essential genome in pathogenic strains such as ST131, the model strain BW25113 was used as a basis for our study. The predicted amino acid sequences of 353 out of the 358 essential genes identified by Goodall et al. [10] were retrieved (Figure 1). The remaining five genes (*ttcC*, *yddL*, *yedN*, *ygeF* and *ygeN*) were labelled as ‘pseudogenes’ or ‘putative proteins’. None of these proteins had been identified as essential in the Keio collection, a systematic collection of single-gene knockout mutations previously used as the gold standard to define the essentialome of *E. coli* BW25113 [11]. Such genes were excluded from further analysis since their classification as essential by Goodall et al. [10] may be an artefact of the methodology employed in the original study.

The sequences of the 353 proteins were compared to those found in *E. coli* O25b:H4-ST131 using a pre-defined cut-off (bitscore ≥ 50 or ≥ 70% sequence identity and ≥ 75% alignment length, see Materials and Methods). All of the 353 essential BW25113 proteins were associated with at least one hit in ST131, apart from YmfE and YmiB (Appendix A). Eleven additional proteins were excluded as they scored below the cut-off threshold, leading to a list of 340 essential and conserved proteins for further analyses. Inspection of the excluded sequences revealed that 10 were prophage-related or uncharacterized proteins. The presence of phages in a bacterial genome is expected to vary with the specific strain history, and this may explain the observed difference between the laboratory strain BW25113 and the hyper-virulent clonal lineage ST131. The essentiality status of some of these phage-related genes (e.g., *relB, ydaS* and *racR*) can be explained through their function as transcriptional regulators, and removal of these proteins allows the phage to activate and subsequently kill the cell.

### 2.2. Similarity to Proteins in Mammalian Hosts

The second step of the analysis aimed at removing *E. coli* targets homologous to the human proteome. A high degree of similarity between the pathogen-specific target and one or more proteins in the host proteome may result in off-target binding of a drug, leading to toxicity and unwanted side effects. The 340 selected essential proteins were therefore compared to the human proteome, leading to 181 proteins fulfilling the same stringent cut-offs as described above (Figure 1, Appendix A).

### 2.3. Similarity to Proteins in Beneficial Taxa of the Gut Microbiota

The next step in the selection pipeline aimed to exclude proteins with high similarity to those found in representatives of the beneficial gut microbiota. Given the complexity and variability of the gut microbiome, we decided to focus on seven taxa containing species previously shown to have beneficial and protective effects on the host [7,8,12,13,14,15,16,17,18]: *Faecalibacterium*, *Prevotella*, *Ruminococcus, Bacteroides, Lactobacillus, Lachnospiraceae* and *Bifidobacterium* (Appendix A). The 181 proteins were blasted against the abovementioned taxa using the same cut-off values as before (Figure 1). As expected, this step was the most selective, leaving just 26 proteins for further analysis (Table 1), and removed all targets of commercially available antibiotics, including FtsI and MrdA (targets of β-lactams), as well as parts of the 30S and 50S ribosome (targets of macrolides, aminoglycosides and tetracyclines) and RNA polymerase (target of rifamycins).

Among the identified 26 proteins, only PheM and TrpL, leader peptides in the Phe tRNA synthetase and Trp biosynthetic operon, respectively, were found to be missing completely in all taxa. The uncharacterized protein YqeL was found to be missing in all except for *Lachnospiraceae* and *Lactobacillus*. SafA (part of the low pH stress response) was found to be missing in *Lachnospiraceae*, *Bifidobacterium* and *Faecalibacterium.* MreD, a rod shape-determining protein was missing from *Bacteroides* and *Bifidobacterium*. Furthermore, WzyE (probable ECA polymerase) lacked hits in *Bifidobacterium* and *Ruminococcus.* Neither LolA, LolB (both involved in lipoprotein transport) nor DnaT (primosomal protein 1) were associated with any hits in *Bifidobacterium*; LptE (part of the lipopolysaccharide (LPS) assembly machinery) and TusE (a sulphur transferase) were missing in *Faecablibacterium*; FtsL (a cell division protein) was missing from *Bacteroides*; and CydX (cytochrome *bd*-I ubiquinol oxidase subunit X) from *Ruminococcus.* All other proteins were associated with hits below the cut-off in all taxa.

Due to the high stringency applied in the above step, potentially valuable targets may have been missed in the selection process. Thus, a second analysis of the microbiota BLAST results was undertaken to find proteins associated with only 10 or less hits over cut-off. Ten proteins (BamD, FabA, LptD, MukB, MukF, SecE, YdfO, YgfZ, YrfF and ZipA) were each found to have been excluded based on bitscore cut-off rather than sequence identity. Thus, these proteins were included in further analyses, leading to 36 proteins as potential *E. coli*-selective targets (Table 1).

### 2.4. Target Essentiality and Conservation in K. pneumoniae

We evaluated presence and essentiality of the selected targets in *K. pneumoniae*, which represents another global priority pathogen closely related to *E. coli*. The essentiality of the 36 proteins was checked against the library generated by Ramage et al. [29], together with conservation of the amino acid sequence as above (Figure 1, Appendix A). Eighteen were found to be essential in both organisms, all displaying high sequence conservation. However, all of the remaining proteins not reported to be essential in *K. pneumoniae* fulfilled the similarity-based selection criteria, except for YqeL, which lacked hits completely (Table 1).

### 2.5. Biological Function of Selected Targets

Of the 36 identified targets (Table 1), 20 were found to share similar biological functions (Figure 2). The largest functional group comprised of eight proteins involved in outer membrane (OM) biogenesis and maintenance (BamD, LptA, LptD, LptE, LolA, LolB, SecE and YciS) (Figure 2). Cell division made up the second largest functional group (MukB, MukE, MukF, FtsB, FtsL, FtsQ and ZipA), which is closely linked to the functional groups involved in cell shape (MreD) and DNA replication (HolD, PriB, DnaT and YgfZ) (Figure 2). Additionally, two members belonging to the functional group involved in transcriptional regulation (PheM and TrpL) were also identified, together with one member involved in translation (TusE). Furthermore, three proteins were involved in biosynthetic pathways (WzyE, FabA and HemD), three in stress response (CydX, IraM and SafA) and two in toxin–antitoxin systems (HipB and HigA) (Figure 2). Finally, five proteins with unknown functions were also identified (YrfF, YdhL, YcaR, YqeL and YdfO).

### 2.6. Target Localization

An essential requirement for developing an efficient antimicrobial drug is target accessibility. This is especially important in Gram-negative bacteria, where the double membrane structure acts as a permeability barrier, efficiently blocking many compounds from accessing intracellular targets. SCL was therefore considered to evaluate protein’s druggability. The SCL for 25 of the 36 selected proteins could be determined using Swiss-Prot, the manually annotated section of UniProtKB (Figure 2, Table 1). For eleven proteins (DnaT, HemD, HigA, HipB, HolD, PheM, PriB, TrpL, YdfO, YdhL and YqeL), no information about SCL was available. Based on this criterion, the four OM-associated proteins (LptD, LptE, LolB and BamD) are promising potential targets, especially LptD, which contains extracellular domains.

### 2.7. Existence of Known Inhibitors

Next, a literature search was conducted to identify previously reported inhibitors of the selected *E. coli*-specific targets. As expected, none of the targets presented in Table 1 are inhibited by commercially available antibiotics. Through analysis of scientific literature, we were able to identify inhibitors targeting a few of the listed targets but, to our knowledge, none have gone beyond laboratory studies: the ZipA/FtsZ interaction has been reported to be inhibited by certain antimicrobial compounds [27,28]; the insect peptide Thanatin blocks LptA [23]; compound IMB-881 blocks the interaction between LptA and LptC [24]; JB-95 inhibits β-barrel proteins including LptD [25]; MAC13243 inhibits LolA [22]; BamD is inhibited by an inhibitory peptide [19] while the compound IMB-H4 has been shown to block BamA–BamD interaction [20]; MukB is inhibited by the small molecules Michellamine B and NSC260594 [26]; and FabA by 3-Decynoyl-NAC [21]. Thanatin has been shown to possess antimicrobial activity against several Gram-negative bacteria beyond *E. coli*, including *K. pneumoniae*, *Salmonella typhimurium* and *Enterobacter cloacae* [23]. IMB-H4 was also able to inhibit growth in *K. pneumoniae*, *P. aeruginosa* and *A. baumannii* [20]. NSC176319 was found to be active against *Staphylococcus aureus,* permeabilized *P. aeruginosa* and *A. baumannii* [26]. JB-95 was reported to have antimicrobial activity against *A. baumannii*, *P. aeruginosa* and *S. aureus* [25], and MAC13243 has been shown to be active against *P. aeruginosa* [22]. With the information provided in this study, some of these inhibitors may represent starting scaffolds for development into pathogen-specific antibacterials. In addition, they might represent useful tools in validating future target-based assays.

### 2.8. Target Structure

Structure-guided drug design is a powerful in silico approach that can rapidly screen millions of compounds for their ability to dock into a desired target, and identified hits can subsequently be tested *in vitro*. Thus, 3D structures at a high enough resolution represent an advantage for the targets identified in this study.

Information retrieved from the Protein Data Bank (PDB) [30] showed that 3D structure at a resolution of <3 Å was found for 18 proteins, >3 Å for 3 proteins and no structure could be found for 13 proteins, while both YrfF and YdfO were associated with a structure but no resolution information was reported in the databases (Table 1).

## 3. Discussion

The originality of the present study lies in the identification of *E. coli*-selective cellular targets that may lead to the discovery of innovative antimicrobial drugs with limited effect on the healthy gut microbiota. Similar in silico studies have previously been conducted for verotoxigenic *E. coli* O157:H7, *K. pneumoniae, Yersinia pseudotuberculosis* and *Enterobacteriaceae* [31,32,33,34,35]. However, these studies were not designed to identify targets with low identity to the corresponding proteins in beneficial taxa residing in the intestinal tract, and some suffered from limitations related to the lack of a well-established essential genome for the target pathogen.

We identified 36 potential drug targets selective for *E. coli* based on protein sequence identity. A large proportion of the identified proteins are functionally related amongst themselves. Within the proteins involved in OM biogenesis, BamD is directly associated with the OM, and is part of the β-barrel assembly machinery (BAM); LptA, LptD and LptE are all part of the lipopolysaccharide transport (LPT) machinery; LolA and LolB are located in the periplasmic space and on the periplasmic side of the OM respectively, both belonging to the lipoprotein transport machinery responsible for delivering OM lipoproteins to the OM assembly machineries (LOL, BAM and LPT) [36]; SecE is part of the SecYEG protein translocation machinery responsible for transporting proteins into the periplasm [37]; and the inner membrane protein YciS (also known as LapA, lipopolysaccharide assembly protein A) is part of the machinery responsible for envelope stress response and regulation of LPS production [38].

Two functional groups responsible for DNA replication (HolD, PriB, DnaT and YgfZ) and cell division (MukB, MukE, MukF, FtsB, FtsL, FtsQ and ZipA) were identified. DNA replication is a tightly controlled mechanism where the DNA Polymerase III holoenzyme is the major replication complex in *E. coli*, and wherein HolD (ψ subunit) makes up parts of the clamp-loading complex [39]. DNA damage can cause this machinery to stall and disassemble on the chromosome, leading to replication failure. To restart replication, the cell must make use of the replication restart primosome, where PriB and DnaT primase is found [40]. YgfZ has been shown to be part of the system regulating chromosomal replication [41]. The MukBEF complex is found only in a subset of γ-proteobacteria and is involved in cell division, making up the sole *E. coli* condensin for chromosome replication, segregation and organization [42]. Further downstream in this process, the transmembrane complex FtsBL is found [43], which together with FtsQ [44] and ZipA, is involved in cell division [45]. In two related processes, MreD is involved in determining cell shape [46] and TusE in translation [47].

Three selected targets, CydX, IraM and SafA, have functions related to stress response. CydX is part of the CydAB cytochrome *bd* oxidase complex involved in aerobic respiration and maintaining the charge across the membrane used for ATP synthesis [48]. IraM is a regulator of σ^S^, the stationary phase sigma factor responsible for controlling expression of a plethora of genes involved in stress response [49]. SafA is a protein that connects the signal transduction between the two-component systems EvgS/EvgA and PhoQ/PhoP in response to acid stress conditions [50].

Four selected targets are involved in biosynthetic pathways. WzyE has been implicated in the assembly of the enterobacterial common antigen [51]; FabA is a protein involved in fatty acid biosynthesis [52]; TrpL is involved in controlling tryptophan biosynthesis [53]; and HemD is a uroporphyrinogen III synthase [54]. The remaining targets include two anti-toxins of the Type II toxin–antitoxin system, HipB and HigA, which counteract the effect of their cognate toxins [55]. One protein involved in transcriptional regulation, PheM, which is responsible for attenuation of phenylalanyl-tRNA synthetase was identified [56]. Finally, no information regarding biological function could be found for the remaining five proteins (YrfF, YcaR, YdhL, YqeL or YdfO), indicating that there is more to discover regarding *E. coli* biology.

When searching for homologues in the seven representative taxa of the healthy gut microbiota, only TrpL and PheM lacked hits in all these groups. However, no information on SCL and 3D structure is available for either of these two proteins. *trpL* is part of the essential tryptophan biosynthesis pathway, the costliest of the amino acid synthesis pathways. All organisms capable of this biosynthesis employ structurally similar proteins, but the organization within the *trp* operon and its regulatory mechanisms vary widely between different organisms. *trpL* functions as an operon leader peptide, which under low levels of uncharged tRNA^Trp^, causes the ribosome to stall during translation of this operon. This leads to the formation of an antiterminator structure, allowing translation to continue [57]. Similarly, *pheM* acts as an operon leader peptide in the phenylalanine biosynthetic pathway and regulates transcription of the *pheMS* operon [56]. Although the mechanisms of control may differ between bacteria, targeting either of these gene products would be challenging as they exert their control on a transcriptional level. Interestingly, though these proteins are not thought to have any functions in *trans*, it has recently been shown that in the α-proteobacterial plant symbiont *Sinorhizobium meliloti*, TrpL can, upon antibiotic exposure, utilize antimicrobial compounds for post-transcriptional regulation of resistance operons, a trait that was further shown to be conserved in other α-proteobacteria [58]. If a similar role can be established for TrpL in *E. coli*, this may be an interesting target for development of helper drugs.

The potential to target an infecting pathogen without affecting the beneficial microbiota is clinically attractive due to increasing concerns regarding the impact of antimicrobial therapy on dysbiosis [59]. The clinical value of such antibiotics would be even higher if their spectrum covered other pathogenic bacterial species. Thus, the sequence conservation and essentiality of the 36 target proteins selective for *E. coli* were analyzed for sequence similarity and essentiality in *K. pneumoniae*, a close relative to *E. coli,* and another major contributor to multidrug-resistant infections worldwide [60]. Although 35 of the 36 proteins were highly conserved in both species, only 18 were found to be essential in this pathogen. This could be due to disparity of information, since the *E. coli* essential genome is well characterized, whereas the essentiality status of genes in *K. pneumoniae* is defined by a single study [29], which may be affected by different methods and conditions to those used to establish the *E. coli* ‘essentialome’.

Due to the double-membrane structure in Gram-negative bacteria, the accessibility of a potential drug target is essential. Proteins located in the OM are therefore considered to be optimal targets [61]. Here, we identified LptD, LptE, LolB and BamD as OM-associated proteins. While LolB and BamD are associated with two separate OM biogenesis pathways, LptD and LptE, together with LptA, all belong to the Lpt pathway responsible for LPS transport. This pathway is integral for cell viability, as Gram-negative bacteria are dependent on it for LPS transport, and could potentially be efficiently targeted without requiring access to the intracellular environment. The Lpt pathway consists of seven essential proteins, of which three have been indicated through this study as appropriate potential drug targets, suggesting that this is a druggable pathway in *E. coli*. Furthermore, LptD also has extracellular domains, indicating that it may be possible to find an inhibitor that interferes with this protein without the need to cross the OM. All three Lpt proteins were found to be associated with hits in the selected gut microbiotal taxa. Based on bitscore, the highest-ranking hit for LptA was found in *Lachnospiraceae* (bitscore = 38, % alignment = 37% and % id = 35%), whereas for LptD and LptE both were found in *Bacteroides* (bitscore = 172, % alignment = 11%, % id = 96%, and bitscore = 35, % alignment = 35% and % id = 34%, respectively).

Notably, the four proteins listed above have excellent ‘druggability’ potential, since in addition to their optimal SCL, they are also essential in *K. pneumoniae* and have known 3D structures. Furthermore, all four proteins displayed low levels of sequence identity to homologous proteins in the selected gut taxa used in this study; in particular, LolB and LptE, which both fell below the cut-offs. Both LptD and BamD were recovered by manual inspection of the proteins excluded. However, both proteins were excluded due to their bitscore, and were only associated with two and four hits, respectively. The selectivity of BamD and LptD can be further evaluated by future in vitro studies using known inhibitors that specifically interfere with these proteins [19,20,25]. Known inhibitors are also available for ZipA [27,28], LptA [23,24], LolA (MAC13243) [22], BamD (peptide and IMB-H4) [19,20], MukB (Michellamine B and NSC176319) [26] and FabA (3-Decynoyl-NAC) [21]. The fact that six inhibitors targeting OM-related processes were found further strengthens the hypothesis that these functions are viable for development of novel antimicrobials.

All in silico studies suffer from drawbacks related to arbitrary cut-off criteria that may lack biological relevance. Too stringent cut-offs potentially exclude valuable drug targets, while too loose criteria may result in an unmanageable list of targets. In the present study, we chose to integrate stringent cut-offs with manual revision in order to minimize exclusion of relevant proteins. Another limitation of this study is that essentiality may differ between in vitro and in vivo conditions. The essential genome established by Goodall et al. [10] was characterized in rich media conditions, and may therefore not include genes that are essential for metabolism inside the host [10,62]. Certain bacterial biosynthetic pathways may be downregulated as the pathogen instead relies on the host to supply nutrients such as amino acids, vitamins and nucleobases [62]. However, targeting biosynthetic pathways involved in maintenance of the cell is likely to represent a target relevant in vitro as well as in vivo [63].

In silico studies such as this are a first but essential step towards the discovery of novel pathogen-targeted antimicrobials. The results of our study provide a starting point towards the identification and development of novel specific antimicrobials targeting *E. coli*. Future wet-lab studies are required to validate the presumptive selective targets identified by the study. High-throughput screens can be applied to find inhibitors interfering with the specific protein targets, e.g. through ‘knock-down’ strains with reduced expression of the target protein. The antimicrobial activity of the identified inhibitors could subsequently be evaluated on a comprehensive strain collection representative of the healthy gut microbiota, or directly on faecal samples using a metagenomics approach in order to assess their selective toxicity towards *E. coli* and other pathogenic *Enterobacteriaceae*.

## 4. Materials and Methods

### 4.1. Protein Essentiality in E. coli

The GenBank record for *E. coli* BW25113 (GenBank: CP009273.1) was downloaded, and the protein sequences from the 358 genes found to be essential by Goodall et al. [10] were extracted. Five genes were removed due to being labeled as pseudogenes (*ttcC*, *yedN*, *ygeF* and *ygeN*) or putative protein (*yddL*).

### 4.2. Protein Homology in E. coli ST131, Humans and Gut Beneficial Taxa

To establish presence of proteins in *E. coli* O25b:H4-ST131 (NCBI:txid941322), NCBI + BLASTp was used to BLAST the 353 protein sequences against this organism using the organism name as Entrez query against the refseq protein BLAST database. Percent alignment was calculated by dividing the length of the hit by the length of the query protein, extracted from the NCBI record. A dual cut-off was used, where hits with percent id ≥ 70% and percent alignment ≥ 75% or bitscore ≥ 50 were excluded. These cut-offs were selected to be equal to, or more stringent than, those used in previous in silico studies [31,32,35,62]. Any hits below the cut-offs were removed, and the remaining were taken on to the next step.

To find human analogues, the NCBI + command line remote BLAST tool was used to BLAST the remaining protein sequences using first the Entrez queries ‘Homo sapiens [Organism]’ against the ‘refseq_proteins’ database, and the output was sorted using the same cut-offs as described above.

Remote BLASTp command line applications were used to assess protein homology in specific gut taxa using the Entrez queries ‘Faecalibacterium [Organism]’, ‘Bacteroides [Organism], ‘Ruminococcus [Organism]’, ‘Prevotella [Organism]’, ‘Lactobacillus [Organism]’, ‘Lachnospiraceae [Organism]’ and ‘Bifidobacterium [Organism]’ using the ‘refseq_proteins’ database. These filters provided 214,344 different entries as listed in Appendix A. The results were downloaded and analyzed using the same methodology and cut-offs as described above.

All hits sorted as above the cut-off in the similarity search against the simplified microbiota were collected and the number of hits for each protein evaluated. A table with the number of hits, together with the information for the highest scoring hit for each protein, was generated. The proteins with fewer than ten hits were manually inspected, and proteins with high-scoring similarity were removed.

### 4.3. Protein Conservation and Essentiality in K. pneumoniae

To assess protein conservation in *K. pneumoniae*, command line applications for BLASTp were used to BLAST remotely against ‘Klebsiella pneumoniae [Organism]’ using the ‘refseq_proteins’ database (Appendix A). The supplementary dataset generated by Ramage et al. [29] was downloaded and used to search for the potential target genes using gene names.

### 4.4. Identification of Inhibitors by Screening Scientific Literature

Scientific literature was screened to identify inhibitors by searching PubMed and Google Scholar using the phrases ‘*Escherichia coli* inhibitor’ or ‘*Escherichia coli* antimicrobial’ in combination with the protein of interest. Abstracts and manuscripts were screened to find inhibitors targeting the protein of interest in *E. coli*.

### 4.5. Protein SCL and 3D Structure

The SCL for each protein was manually checked by querying the UniProt/SwissProt database, and retrieving the information found under ‘Subcellular Localisation’.

Information about 3D structure for proteins in *E. coli* K-12 was manually retrieved through the UniProt/SwissProt entries for each protein individually. The PDB accession number, molecules in complex and resolution were recorded.

## Figures and Tables

**Figure 1 antibiotics-10-00632-f001:**
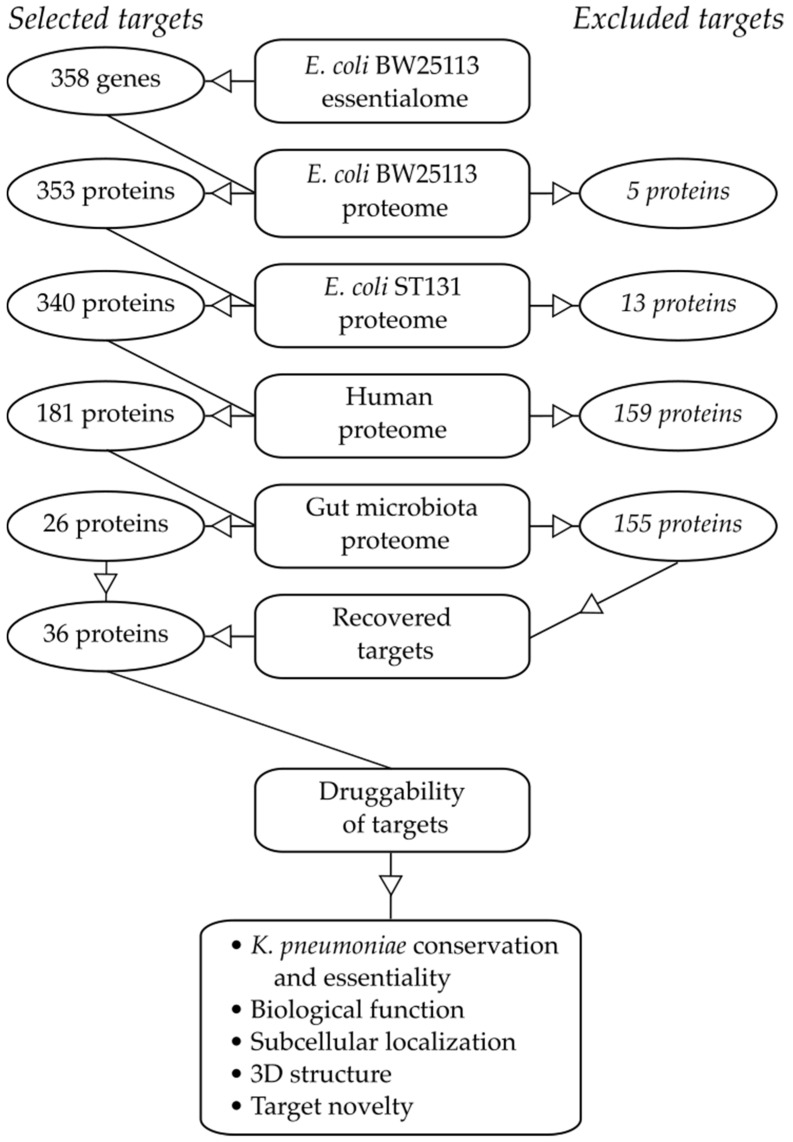
Flow diagram showing the different steps of the selection procedure used to identify novel selective antimicrobial drug targets against *E. coli*.

**Figure 2 antibiotics-10-00632-f002:**
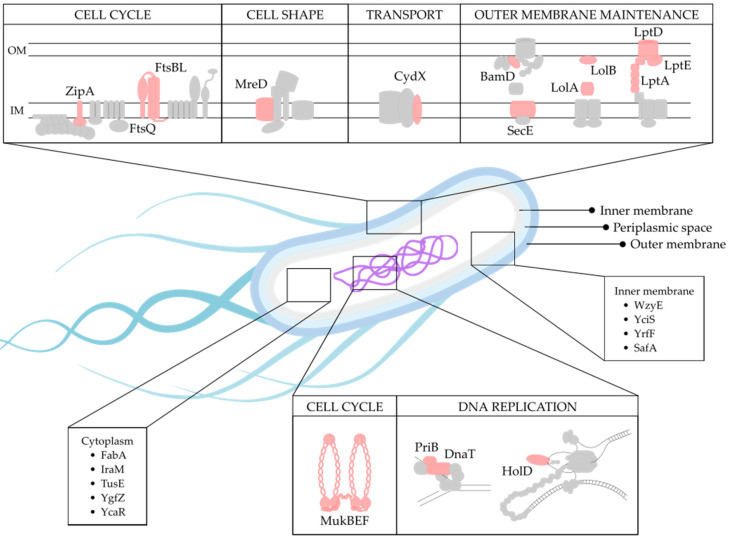
Function and SCL of the identified protein targets. The identified targets are drawn in pink, with surrounding proteins in gray. The proteins drawn are part of various cellular processes, including cell cycle (ZipA, FtsB, FtsL, FtsQ, MukB, MukE and MukF), cell shape (MreD), transport (CydX), outer membrane processes (BamD, SecE, LolA, LolB, LptA, LptD and LptE) and DNA replication (DnaT, PriB and HolD). Other inner membrane proteins are WzyE, YciS, YrfF and SafA, and cytoplasmic proteins are FabA, IraM, TusE, YgfZ and YcaR. Proteins without a determined SCL are not shown: HemD, TrpL, HigA, HipB, PheM, YdfO, YdhL and YqeL.

**Table 1 antibiotics-10-00632-t001:** Gene name, function, PDB accession number, crystal structure and resolution according to RCSB PDB, SCL according to Swissprot, available inhibitors, sequence conservation and essentiality in *K. pneumoniae* for the 36 protein targets identified by this study. N/A = not available, N/D = not determined.

Gene	Protein Function	PDB Accession No.	Crystal Structure and Resolution	SCL	Known Inhibitor	Conservation and Essentiality in *K. pneumoniae*			
						Alignment	aa Identity	Bitscore	Essentiality Status
*bamD*	Outer membrane protein assembly factor BamD	P0AC02	5D0O 2.90 Å	Cell outer membrane; Lipid anchor	Inhibitory peptide [19], IMB-H4 [20]	100%	94%	479	Essential
*cydX*	Cytochrome *bd*-I ubiquinol oxidase subunit X	P56100	6RKO 2.68 Å	Cell inner membrane; Single-pass membrane protein	N/A	100%	100%	77	N/A
*dnaT*	Primosomal protein 1	P0A8J2	4OU6 1.96 Å	N/D	N/A	100%	76%	285	N/A
*fabA*	3-hydroxydecanoyl-[acyl-carrier-protein] dehydratase	P0A6Q3	4KEH 2Å	Cytoplasm	3-Decynoyl-NAC [21]	100%	99%	349	Essential
*ftsB*	Cell division protein FtsB	P0A6S5	4IFF 2.30 Å	Cell inner membrane	N/A	84%	100%	181	Essential
*ftsL*	Cell division protein FtsL	P0AEN4	N/D	Cell inner membrane	N/A	100%	96%	238	Essential
*ftsQ*	Cell division protein FtsQ	P06136	2VH1 2.70 Å	Cell inner membrane; Single-pass type II membrane protein	N/A	96%	89%	486	Essential
*hemD*	Uroporphyrinogen-III synthase	P09126	N/D	N/D	N/A	100%	99%	497	Essential
*higA*	Antitoxin HigA	P67701	6JQ4 2 Å	N/D	N/A	100%	99%	279	N/A
*hipB*	Antitoxin HipB	P23873	2WIU 2.35 Å	N/D	N/A	100%	98%	176	Essential
*holD*	DNA polymerase III subunit psi	P28632	3SXU 1.85 Å	N/D	N/A	84%	98%	224	N/A
*iraM*	Anti-adapter protein IraM	P75987	N/D	Cytoplasm	N/A	99%	62%	149	N/A
*lolA*	Outer membrane lipoprotein carrier protein	P61316	1IWL 1.56 Å	Periplasm	MAC13243 [22]	100%	94%	399	Essential
*lolB*	Outer membrane lipoprotein LolB	P61320	1IWM 1.90 Å	Cell outer membrane; Lipid anchor	N/A	100%	100%	426	Essential
*lptA*	Lipopolysaccharide export system protein LptA	P0ADV1	2R19 2.16 Å	Periplasm	Thanatin [23], Compound IMB-881 [24]	100%	85%	328	Essential
*lptD*	LPS assembly protein LptD	P31554	4RHB 3.35 Å	Cell outer membrane	Inhibitory peptide JB-95 [25]	100%	84%	1407	Essential
*lptE*	LPS assembly lipoprotein LptE	P0ADC1	4RHB 3.35 Å	Cell outer membrane	N/A	102%	72%	295	Essential
*mreD*	Rod shape-determining protein MreD	P0ABH4	N/D	Cell inner membrane; Multi-pass membrane protein	N/A	100%	100%	310	N/A
*mukB*	Chromosome partition protein MukB	P22523	1QHL 2.20 Å	Nucleoid	NSC260594, Michellamine B [26]	100%	92%	2774	Essential
*mukE*	Chromosome partition protein MukE	P22524	3EUH 2.90 Å	Nucleoid	N/A	100%	95%	458	Essential
*mukF*	Chromosome partition protein MukF	P60293	3EUH 2.90 Å	Nucleoid	N/A	100%	97%	878	Essential
*pheM*	Phenylalanine tRNA ligase operon leader peptide	P0AD74	N/D	N/D	N/A	100%	100%	30	N/A
*priB*	Primosomal replication protein N	P07013	5WQV 1.97 Å	N/D	N/A	100%	93%	205	N/A
*safA*	Two-component-system connector protein SafA	P76136	N/D	Cell inner membrane; Single-pass type II membrane protein	N/A	98%	100%	127	N/A
*secE*	Protein translocase subunit SecE	P0AG96	5GAE 3.33 Å	Cell inner membrane; Multi-pass membrane protein	N/A	100%	98%	240	Essential
*trpL*	trp operon leader peptide	P0AD92	N/D	N/D	N/A	100%	100%	32	N/A
*tusE*	Sulfurtransferase TusE	P0AB18	N/D	Cytoplasm	N/A	100%	90%	204	Essential
*wzyE*	Probable ECA polymerase	P27835	N/D	Cell inner membrane	N/A	100%	87%	744	N/A
*ycaR*	UPF0434 protein YcaR	P0AAZ7	N/D	Cytoplasm	N/A	100%	100%	125	N/A
*yciS*	Lipopolysaccharide assembly protein A	P0ACV4	N/D	Cell inner membrane	N/A	100%	87%	177	N/A
*ydfO*	Uncharacterized protein YdfO	P76156	2HH8 N/D	N/D	N/A	100%	79%	228	N/A
*ydhL*	Uncharacterized protein YdhL	P64474	N/D	N/D	N/A	96%	98%	152	N/A
*ygfZ*	tRNA-modifying protein YgfZ	P0ADE8	1VLY 1.30 Å	Cytoplasm	N/A	100%	84%	573	N/A
*yqeL*	Uncharacterized protein YqeL	C1P613	N/D	N/D	N/A	N/A	N/A	N/A	N/A
*yrfF*	Putative membrane protein IgaA homolog	P45800	4UZM N/D	Cell inner membrane; Multi-pass membrane protein	N/A	100%	73%	1089	N/A
*zipA*	Cell division protein ZipA	P77173	1F46 1.50 Å	Cell inner membrane; Single-pass type I membrane protein	Antimicrobial compounds [27,28]	101%	98%	646	Essential

## Data Availability

The data presented in this study are available in the article and Appendix A.

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
