# Peer review of "In Silico Prediction and Prioritization of Novel Selective Antimicrobial Drug Targets in Escherichia coli"

_antibiotics, 2021, doi:10.3390/antibiotics10060632_

Round 1
Reviewer 1 Report
In the manuscript entitled “In silico prediction and prioritisation of novel selective antimicrobial drug targets in Escherichia coli”, Frida Svanberg Frisinger and colleagues report on a bioinformatics analysis to find possible suitable targets to guide future antibiotic discovery and development. Even if the paper presents interesting points such the removal of potential candidates homologous to human proteins or to targets present in beneficial taxa of the human holobiont, it lacks any experimental, “wet lab” data, which would have given a huge relevance to the study. In addition, it also suffers of some major problems, mostly related to bioinformatics approaches, as detailed below.
General points:
- The choice of an Escherichia coli strain to start the analysis is not optimal in my opinion. This species is in fact a constitutive species of the healthy human gut microbiota, therefore molecules against all the “druggable” targets presented in this study could potentially interfere with the homeostasis of E.coli in the intestine. Authors should filter their targets to find out which are present only in pathogenic E.coli strains but not in the “beneficial” strains of this species.
- The term homology is a qualitative and not a quantitative feature in bioinformatics (i. e. two sequences are either homologous or not, and homology cannot be quantified). Therefore, terms such as “similarity” or “identity” in place of “homology” should be used in all the points where quantitative comparison is reported (e.g. line 13, line 65, etc).
-In BLAST, E value is dependent on database size and, therefore, ideally results obtained by different BLAST searches using different databases cannot be compared. The bitscore index, which is independent on both the DB size and the scoring matrices employed to evaluate, e.g, protein alignments, must be used instead of E value.
-When using the nr database a lot of genomes (i.e. those deposited in the WGS division, which account for the majority of sequenced genomes) are not searched for. This could lead to results that does not take into account the real biological variability actually present in public DNA databases.
- Even if the paucity of new antimicrobials is a major contemporary problem, there are several novel antibiotics (e.g. tigecycline, ceftolozane–tazobactam, ceftazidime–avibactam, meropenem–vaborbactam, plazomicin, imipenem–relebactam, cefiderocol, etc) that are now available for the effective treatments of Gram-negative infections. These options should be briefly discussed in the introduction section to provide the reader a more realistic view of the last drugs available for therapy.
-When the Authors searched their targets in members of the Klebsiella pneumoniae genus, they focused their search on a single representative (i.e. KPNIH1, line 143). Even if this strain is a member of ST258, a major clone for the pandemic dissemination of MDR Klebsiella pneumoniae, the analysis of the presence/conservation of “druggable” targets had to be expanded to take into account at least all major MDR and/or hypervirulent clones of Klebsiella pneumoniae (e.g. strains of clonal group 258, including ST512, ST258 clade I and ST258 clade II, ST101, ST15, ST307, ST23, etc ). Moreover, the presence of a given target in a specific isolate doesn’t means that it is present in all members of a given clonal group. Please, see Wyres et al., Nat. Rev. Microbiol. 2020, to expand the list of K.pneumoniae strains to be selected for this search.
Minor points:
-line 22. Enterobacteriaceae should be italicized. A similar correction should be done for “Escherichia coli” at line 27. Please, double-check the manuscript to report species/bacterial families always in italic characters.
-lines 76-78. Please, revise this sentence. Is there a superfluous “and”?
-line 84. “15 proteins” or 16, as shown in Figure 1?
-lines 106-107. Please, revise this sentence. “have previously been” should be “previously”?
-line 111. “31 proteins” or 30, as shown in Figure 1?
-line 401 “Bifidobacterium[Organism]” instead of “fidobacterium[Organism]”?
Reference section:
Please, double-check that species and gene names are reported in italic characters
Author Response
Point-by-point Responses to reviewer 1 of manuscript ID Antibiotics 1182105
In the manuscript entitled “In silico prediction and prioritisation of novel selective antimicrobial drug targets in Escherichia coli”, Frida Svanberg Frisinger and colleagues report on a bioinformatics analysis to find possible suitable targets to guide future antibiotic discovery and development. Even if the paper presents interesting points such the removal of potential candidates homologous to human proteins or to targets present in beneficial taxa of the human holobiont, it lacks any experimental, “wet lab” data, which would have given a huge relevance to the study. In addition, it also suffers of some major problems, mostly related to bioinformatics approaches, as detailed below.
Response: Regarding the lack of “wet lab” data, validation of the results of our in silico study requires identification of at least one compound that interferes with one of the identified targets. We are currently performing a drug screen in collaboration with a private company to identify compounds targeting LptD. However, identification and characterization of such compounds require time, resources and efforts that justify publication of a separate manuscript to describe this follow-up study.
We thank the reviewer for the comments regarding the bioinformatics approach, which have been addressed in the revised manuscript.
General points:
- The choice of an Escherichia coli strain to start the analysis is not optimal in my opinion. This species is in fact a constitutive species of the healthy human gut microbiota, therefore molecules against all the “druggable” targets presented in this study could potentially interfere with the homeostasis of E.coli in the intestine. Authors should filter their targets to find out which are present only in pathogenic E.coli strains but not in the “beneficial” strains of this species.
Response: Your idea of identifying targets that are specific to hyper-virulent E. coli clones is interesting but goes beyond the objective of our study, which was to identify targets with decreased risk of causing dysbiosis. For different reasons we think that our choice to use E. coli as a target is viable. The distinction between pathogenic and beneficial E. coli is rather artificial and subjective since any E. coli can cause infection. For example, various studies have shown that UTI, which is the most common infection associated with E. coli, is often caused by the most prevalent strain in the faecal microbiota of the patient [1,2]. Moreover, dysbiosis is associated with an overall increase in the proportion of Enterobacteriaceae and E. coli regardless of the pathogenic potential of the strains that are selected by antibiotic exposure.
- The term homology is a qualitative and not a quantitative feature in bioinformatics (i. e. two sequences are either homologous or not, and homology cannot be quantified). Therefore, terms such as “similarity” or “identity” in place of “homology” should be used in all the points where quantitative comparison is reported (e.g. line 13, line 65, etc).
Response: We agree. This has now been corrected throughout the manuscript.
-In BLAST, E value is dependent on database size and, therefore, ideally results obtained by different BLAST searches using different databases cannot be compared. The bitscore index, which is independent on both the DB size and the scoring matrices employed to evaluate, e.g, protein alignments, must be used instead of E value.
Response: This is a good point, thank you. E-value has now been replaced by bitscore as the selection criterion.
-When using the nr database a lot of genomes (i.e. those deposited in the WGS division, which account for the majority of sequenced genomes) are not searched for. This could lead to results that does not take into account the real biological variability actually present in public DNA databases.
Response: This is another valid point. After having spoken to NCBI support, we have now changed this to ‘refseq proteins’. However, the support team at NCBI also pointed out that both databases contain proteins derived from WGS, especially for bacteria as the vast majority of genomes now derive from WGS. This change, together with the change from e-value to bitscore led to the inclusion of 36 proteins in the final target list, as compared to the previous 37 targets. Five proteins were excluded (HolA, LptF, PssA, TonB and YobI), and 4 new proteins were included (FabA, TusE, YdfO and YqeL) after this change.
- Even if the paucity of new antimicrobials is a major contemporary problem, there are several novel antibiotics (e.g. tigecycline, ceftolozane–tazobactam, ceftazidime–avibactam, meropenem–vaborbactam, plazomicin, imipenem–relebactam, cefiderocol, etc) that are now available for the effective treatments of Gram-negative infections. These options should be briefly discussed in the introduction section to provide the reader a more realistic view of the last drugs available for therapy.
Response: We added this information to the introduction (lines 32-35)
-When the Authors searched their targets in members of the Klebsiella pneumoniae genus, they focused their search on a single representative (i.e. KPNIH1, line 143). Even if this strain is a member of ST258, a major clone for the pandemic dissemination of MDR Klebsiella pneumoniae, the analysis of the presence/conservation of “druggable” targets had to be expanded to take into account at least all major MDR and/or hypervirulent clones of Klebsiella pneumoniae (e.g. strains of clonal group 258, including ST512, ST258 clade I and ST258 clade II, ST101, ST15, ST307, ST23, etc ). Moreover, the presence of a given target in a specific isolate doesn’t means that it is present in all members of a given clonal group. Please, see Wyres et al., Nat. Rev. Microbiol. 2020, to expand the list of K.pneumoniae strains to be selected for this search.
Response: We agree, and the search has now been extended to include K. pneumoniae as a species.
Minor points:
-line 22. Enterobacteriaceae should be italicized. A similar correction should be done for “Escherichia coli” at line 27. Please, double-check the manuscript to report species/bacterial families always in italic characters.
Response: we corrected this mistake throughout the manuscript.
-lines 76-78. Please, revise this sentence. Is there a superfluous “and”?
Response: this sentence was revised as suggested.
-line 84. “15 proteins” or 16, as shown in Figure 1?
Response: this inconsistency was corrected. We excluded 11 proteins that were either absent (n=2) or scored below the threshold (n=11) following comparison to the genome of ST131.
-lines 106-107. Please, revise this sentence. “have previously been” should be “previously”?
Response: the sentence was revised as suggested.
-line 111. “31 proteins” or 30, as shown in Figure 1?
Response: this inconsistency was corrected (36 proteins after revision of the analysis based on your comments).
-line 401 “Bifidobacterium[Organism]” instead of “fidobacterium[Organism]”?
Response: this typo was corrected.
Reference section:
Please, double-check that species and gene names are reported in italic characters
Response: done
References
- Nielsen, K.L.; Dynesen, P.; Larsen, P.; Frimodt-Møller, N. Faecal Escherichia coli from patients with E. coli urinary tract infection and healthy controls who have never had a urinary tract infection. J. Med. Microbiol. 2014, 63, 582–589, doi:10.1099/jmm.0.068783-0.
- Moreno, E.; Andreu, A.; Pigrau, C.; Kuskowski, M.A.; Johnson, J.R.; Prats, G. Relationship between Escherichia coli strains causing acute cystitis in women and the fecal E. coli population of the host. J. Clin. Microbiol. 2008, 46, 2529–2534, doi:10.1128/JCM.00813-08.

Reviewer 2 Report
Interesting and well organised paper. Line 333 the word "intercellular" should be replaced with "intracellular". It would be useful to mention in the conclusions the ultimate requirement to test the toxicity of these lead compounds in mammalian and human cells in vitro, and ultimately in vivo. Indeed it would be advisable to test toxicity in mammalian systems as soon as possible (in fact, it might be possible to do this computationally by matching the 37 lead compounds against databases of toxic compounds, or chemical structures associated with mammalian toxicity, as a final "filter" on the selected compounds). This should be mentioned. This crucial extension of the work could be the basis for the follow-up publication.
Reviewer 3 Report
I read with interest this manuscript that reports the identification of specific cellular targets in a E,coli pathogen model that may lead to the discovery of novel antimicrobials with a low impact in the recognized beneficial bacteria from gut microbiota. This is an innovative, original research and will contribute for the discovery of new antimicrobials targeting specifically pathogens, and not all bacteria. The rational of the study is well done. Results are clear, showing tables and figures that contribute to the understanding of the text; results are presented in detail, namely the function of all the proteins in study, and with supplementary material. The results are well discussed and convincing. One the points that I would comment, authors are also aware of the limitations of in silico studies, and discuss this limitation, inherent to the methodology.
Overall, I have no further comments except a little suggestion. Since this work will be of the interest of scientists working with microbiota, I would insert it in the keywords. Two of them are already in the title.
References should be edited for italics, etc.
Round 2
Reviewer 1 Report
The revised version of this manuscript presents significant improvements and I have no further comments.